# Effect of safety and security equipment on patient and visitor violence towards nurses in multiple public hospitals of China during the COVID-19 pandemic: a retrospective, difference-in-difference analysis

Yanzhen Hu ,[1] Ju Huang ,[2] Dan Zhao,[1] Cheng Zhang,[2] Jinghua Xia ,[1] Xue-mei Lu[1]

YH and JH are joint first authors.

[1]Department of Nursing, Beijing Jishuitan Hospital, Capital Medical University, Beijing, Beijing, China
[2]Institute of Medical Information, Chinese Academy of Medical Sciences & Peking Union Medical College, Beijing, China

**Correspondence to**
Professor Xue-mei Lu;
jstlxm@163.com

## ABSTRACT

**Objectives** This study aimed to analyse whether safety and security equipment decreased patient and visitor violence (PVV) towards nurses in the COVID-19 period and quantify to what extent safety and security equipment affects PVV.

**Design** Controlled before and after study and difference-in-difference (DID) analysis.

**Setting** A large hospital medical group, consisting of three public tertiary teaching hospitals, namely, Xinjiekou Branch, Huilongguan Branch and Xinlongze Branch of Beijing Jishuitan Hospital, located in the west and north parts of Beijing, China.

**Participants** A panel of nine departments recruited using two-step sampling method, administered online surveys in 2021 and 2022. A total of 632 eligible nurses participated in the survey in 2021 and 725 eligible nurses in 2022.

**Measures** We assessed impacts of the safety and security equipment on the PVV. The policy had been enacted in June 2020, and the corresponding measures were established after mid-December 2020, and therefore, we use a DID design to evaluate changes in nurses' PVV incidence. Departments are classified as either department installed or non installed, and nurses are classified based on their department.

**Results** Within the treatment group, the incidence of physical PVV significantly decreased from 13.8% in 2020 to 2.0% in 2021. In the control group, the incidence of physical PVV increased from 0.6% in 2020 to 2.7% in 2021. The application of the safety and security equipment decreased the incidence of physical PVV by 13.93% (95% CI: −23.52% to −4.34%). In contrast, no difference was observed between the treatment and control groups for the incidence of psychological PVV (6.23%, 95% CI: −11.56% to 24.02%) and overall PVV (0.88, 95% CI: −20.90% to 22.66%).

**Conclusion** The safety and security equipment reduced the incidence of physical PVV towards nurses. For hospital managers in public hospitals, longer-term strategies roadmap for PVV prevention measures are needed to create a more supportive work environment in employees.

## STRENGTHS AND LIMITATIONS OF THIS STUDY

⇒ This study is the first to quantitatively assess the impacts of the safety and security equipment on patient and visitor violence (PVV) prevention in nurses.

⇒ The safety and security equipment of public hospitals has been improved in China during the past 2 years, and the impact of safe and security equipment on PVV can be visualised.

⇒ The study collected data from three public hospitals with independent operation and varied geographic in a large hospital medical group.

⇒ Using the difference-in-difference model to verify causal correlations, this study empirically ascertains the influence of safety and security measures on PVV at the department level and quantifies its magnitude.

## INTRODUCTION

Violence and harassment against health workers (HWs) have been increasing, especially as the world fights COVID-19, which has been ongoing for 3 years, putting HWs under tremendous anxiety, burnout and stress in hospitals.[1 2] Workplace violence (WPV) is a serious threat to HWs; the Bureau of Labor Statistics notes that of 15 980 private industry workers who experienced trauma from non-fatal WPV, 69% worked in healthcare.[3] The International Labour Organization defines WPV as incidents in which staff are abused, threatened or assaulted in their work, which is divided into physical and psychological violence.[4] The most prevalent type of WPV in healthcare is patient and visitor violence (PVV), which can lead to physical injury (eg, beating, kicking, slapping, stabbing, shooting, pushing, biting and pinching, among others)

and psychological harm (eg, verbal abuse, bullying/mobbing, harassment and threats).[5]

Globally, there have been reports of frontline HWs, especially nurses, being attacked by perpetrators because of fear, panic, misplaced anger and misinformation during the COVID-19 pandemic.[1] According to the International Committee of the Red Cross, more than 600 cases of violence against HWs occurred in the first 6 months of the pandemic in the world, but cases that have been reported are just only a tip of the iceberg.[6] Byon *et al*[7] reported that 44.4% and 67.8% of nurses experienced physical and non-physical violence, respectively, during a 5-month period early in the COVID-19 pandemic in the USA. In China, many nursing staff, particularly experienced nurses, have worked on nasopharyngeal swab collection and other epidemy-related tasks, which may have resulted in nurses working long hours, insufficient rest and constant overtime staff shortages. The shortage may aggravate the PVV.[8] In addition, anxiety, work pressure and inadequate infection prevention and control (IPC) measures could have led to PVV during the early stage of COVID-19.[9]

With the improvement of IPC measures for frontline HWs in COVID-19, there are several countries which have developed and begun to implement coordinated strategies such as a 'zero tolerance zone' for the prevention of WPV in nurses.[10 11] The most basic element to protect against violence in the workplace is to have safe working conditions; thus, environmental controls are a key element of occupational violence and aggression prevention and management programmes in healthcare worldwide, consisting of an increased presence of hospital security staff and equipment.[12] In China, the government approved the Medicine and Health Promotion Law, which took effect in June 2020 and included provisions to prevent PVV against medical workers. It emphasises that no person or organisation may threaten or endanger the personal safety of medical personnel or infringe on their personal dignity.[6] Under this policy, safety and security measures at the organisational and environmental levels were implemented for HWs by tertiary hospitals in mid-December 2020 in Beijing, including installing security cameras and alarm buttons, providing security staff with walkie-talkies and increasing the numbers of metal detectors and security channels.[13]

The implementation of workplace hazard prevention and control measures associated with violence against HWs has been studied worldwide for years, and the measures can be divided into two types: training plans (for internal factors of PVV, perpetrator or recipient characteristics) and organisational and environmental plans (for external factors, policies and the workplace environment).[14] Currently, there are a few studies that have focused on external factors. Safety and security equipment as environmental prevention strategies attracts little attention. Although research on the direct effects of safety and security equipment on the decrease in PVV during COVID-19 is limited, several positive research results were reported. A cross-sectional study by Henry Ford Hospital in Michigan evaluated installed metal detectors and found that they reduced violent crimes by 65% over 18 months. The metal detectors relieve the pressure on healthcare workers and decrease PVV.[15] Blando *et al*[16] also found in a cross-sectional study that metal detectors can support weapon identification and removal, enhancing safety in healthcare facilities. Yet, owing to the limitations of the cross-sectional design in the existing research, the causal associations between safety and security equipment and PVV remain unknown.

Considering the scant evidence evaluating the efficacy of safety and security equipment on PVV prevention, particularly regarding nurses, we used panel data from 2020 to 2021, 2 years before and after the implementation of safety and security equipment, respectively, which adopted a difference-in-difference (DID) design, to examine the effects of safety and security equipment on PVV against nurses. The PVV incidents towards nurses during the COVID-19 pandemic and its causal correlations with the implementation of safety and security measures are expected to be found in the study. In addition, the DID model used in this study could provide quantitative data to existing knowledge, quantifying the reduction rate in PVV with the implementation of safety and security measures. It offers a more objective data reference for the addition of new security measures in the next phase, furthering the goal of zero tolerance for physical violence. Such findings offer valuable guidance for hospitals in other developing countries confronting similar situation. Moreover, our research augments evidence-based practices by providing empirical data on the efficacy of safety equipment.

## METHODS
### Study design and data collection
The data were collected from a panel of departments in three hospitals located on the west and north of Beijing, within a large, public tertiary hospital group. The hospital provides healthcare services nationwide, with 1600 beds, 3509 employees and a yearly outpatient amount that reaches 2.24 million visits. The inclusion criteria for individual participants were full-time employees with qualification certificates and a willingness to answer the survey. The exclusion criterion was being on leave for more than a month in their department. The sample size was calculated using the double population proportion formula in PASS software based on the following assumptions: the prevalence of physical PVV from a previous study as 23%,[17] 80% power to detect a significant difference of 0.05, an effect size of 0.1 and an allocation ratio of 1:3 between two groups. Finally, after accounting for a 10% non-response rate, the sample size of this study minimum yielded 517 respondents each year (120 for the treatment group and 397 for the comparison group).

A two-step sampling method was used to recruit respondents (figure 1). First, the department-level samples

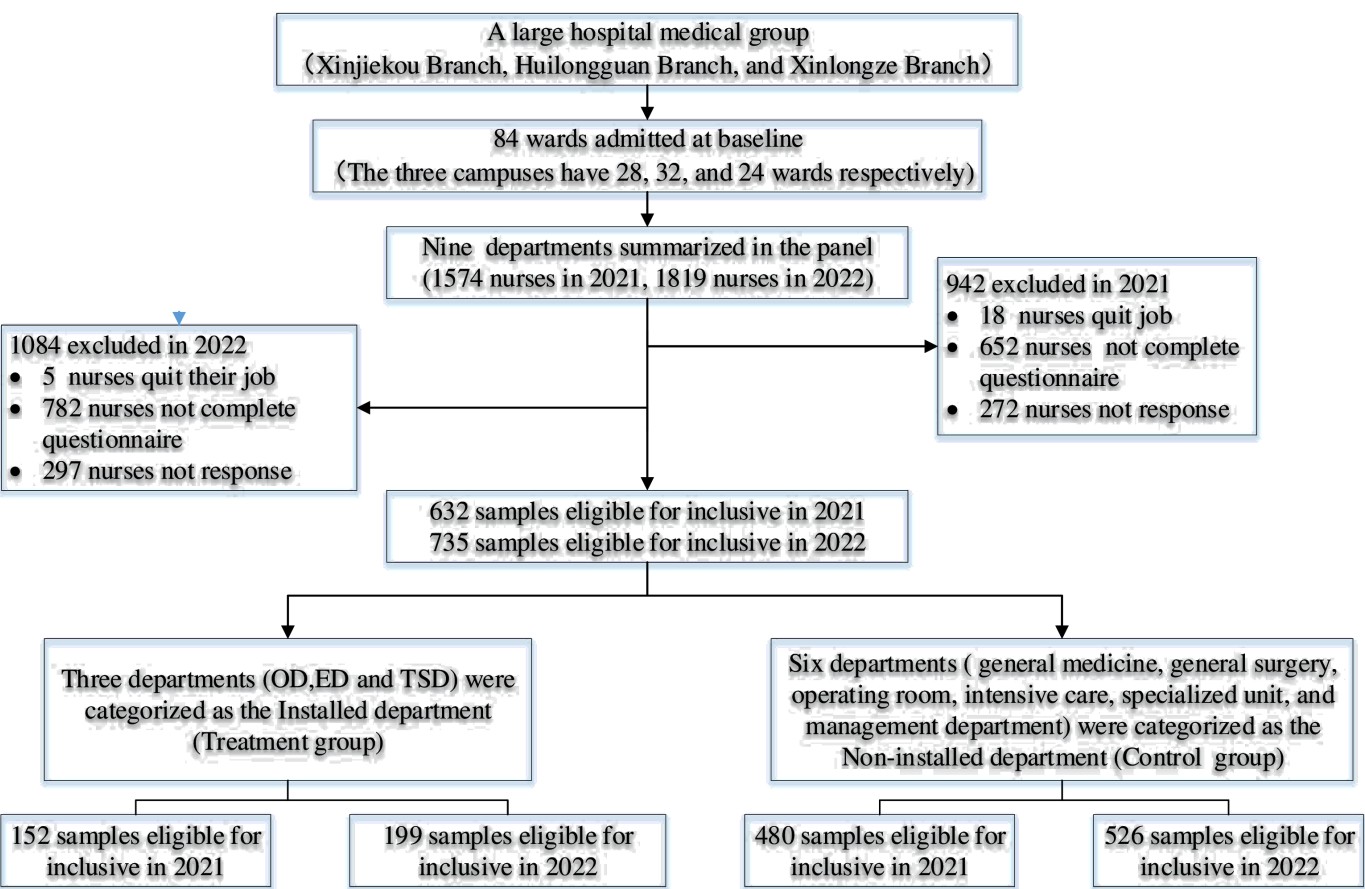

**Figure 1** Sample flow diagram. OD, outpatient department; ED, emergency department; TSD, technical services department.

were selected, which covered all major types of wards in the three hospitals. The final department-level sample comprised nine departments, including the outpatient department (OD), emergency department (ED) and technical services department (TSD) and general medicine, general surgery, operating room, intensive care, specialised unit and management department, and at least two wards were included for each department. Then convenience sampling was performed to collect an individual-level sample from each department. The number of nurses in each selected department was determined and used as the sampling frame. The total sample size was allocated based on the sampling frame, resulting in 10–280 individual participants in each department.

According to the Medicine and Health Promotion Law, based on the epidemiological characteristics of PVV, the hospital administration strategically selected three departments—OD, ED and TSD—for the implementation of safety and security measures. Therefore, the participants from departments that installed the safety and security equipment at the end of 2020 were treatment groups (OD, ED and TSD), and the participants from departments with no safety and security equipment were control groups: general medicine, general surgery, operating room, intensive care, specialised unit and management department. The baseline data for 2020 were collected in January 2021, and we followed up with the selected departments that previously consented to participate in

the panel for data of 2021 in January 2022. Ultimately, a total of 632 eligible nurses participated in the survey 2021 to report the PVV in 2020, of whom 152 were from the treatment group and 480 were from the control group. In survey 2022, 725 eligible nurses participated to report the PVV in 2021, of whom 199 were from the treatment group and 526 were from the control group.

All surveys were administered online by using Wenjuanxing forms, which is the largest computer programme for conducting online surveys in China to help researchers collect data without face-to-face meetings by sending online links to nurses through WeChat. According to the set rules, all items had to be completed before submission, and there were no missing values. Therefore, the forms that have the function of checking the questionnaire automatically to ensure all questions were answered.[18 19]

### Measures

The post-period for this analysis is defined as 2021. While the Medicine and Health Promotion Law was signed in June 2020, implementation was not instantaneous. The safety and security equipment was installed in the hospital in December 2020 and identified by the hospital security officer. Metal detectors (SM-SG-EDD-5000, Sinomis) and security channel (ISD-SC6550D-4YH, Sinomis) were set up at entrances in the outpatient building and emergency building to screen individuals for any potentially

harmful items that could jeopardise the safety of medical personnel. Security cameras (IVS3800XF36 V5, Huawei) were installed at multiple corners covering the entire building, providing constant surveillance and aiding in the identification of potential risks or incidents; Alarm buttons (NNV-30, LonBon) were installed under the triage table in ED and reception in OD, allowing immediate and discreet alerting of security personnel in case of emergencies or threatening situations. About 70 security personnel, each with a walkie-talkie (MTP3150, Xinzhiheng), facilitated swift communication and coordination among them, enabling quick responses to any security issues. These measures collectively aimed to fortify the safety and security infrastructure within the hospitals, proactively deterring and addressing potential threats against the medical staff's safety and personal dignity.

The main outcomes were measured by the structured Chinese revised version of the Workplace Violence in the Health Sector - Country Case Study Research questionnaire, including physical PVV, psychological PVV and overall PVV. The English version was developed in 2003 by the International Council of Nurses, WHO and Public Services International.[20] This questionnaire was revised based on the context of Chinese township hospitals and has been widely used in Chinese public hospitals.[21] The International Questionnaire-C-R includes the following four categories: demographic and workplace data, physical violence within the past 12 months, psychological WPV (verbal abuse, bullying/mobbing, sexual harassment and racial harassment) within the past 12 months and perception of effective strategies for PVV prevention. The reliability and validity of the questionnaire have been tested in studies involving experts from different areas of health in China. Cronbach's alpha was 0.828.[22]

### Statistical analysis

We first present descriptive statistics of our sample; differences in PVV rates were compared using Pearson's $\chi^2$ test. Next, to evaluate the effect of safety and security equipment on the incidence of PVV, the DID model was used to assess the changes of PVV incidence in nurses that occurred pretreatment versus post-treatment. To obtain the significance level of the DID estimate, we adopted a regression model of the general form[23]

$$y_{it} = \alpha + \beta \mathit{intervention}_i + u_i + v_{vt} + \epsilon_{it}$$

where the subscript i indicates department, t indicates year and $y_{it}$ denotes the outcome variable on the incidence of PVV varying across departments and years. The $\mathit{intervention}_i$ is the key explanatory variable of interest, representing the impact of safety and security equipment on the incidence of PVV. $u_i$ is the fixed effect of the control and treatment groups (1 for the department with complete safety and security equipment and 0 for the department without the equipment); $v_t$ is the fixed effect of the pretreatment and post-treatment periods (1 for 2021 and 0 for 2020).

Finally, we controlled the individual-level variables, including age, marital status, job position, work experience, work in shift, work between 6:00 PM and 7:00 AM, interacting with patients/clients and the number of staff in the same work setting. All data analyses were conducted using Stata 16.0. The results were considered statistically significant at $p < 0.05$.

### Patient and public involvement

To engage nurses in the design of this study, we conducted preliminary online communications with them. Throughout the recruitment and factual stages of the study, they were given the autonomy to decide their participation. We assured them that their decisions would not in any way impact their primary job responsibilities. This approach ensured that only voluntary participants were included, honouring their autonomy. We made concerted efforts to mitigate potential burdens or discomfort by maintaining the brevity of the online questionnaire. All participants in the survey were provided written information regarding the objective, context and voluntary nature of their involvement in the study. The anonymity of all participants and the confidentiality of their responses were ensured. This study was approved by the Ethical Review Committee of the Chinese Academy of Medical Sciences (IMICAMS/8/22/HREC).

### RESULTS

The sociological characteristics of the treatment and control groups in 2020 and 2021 are presented in table 1. A total of 1357 nurses (632 in 2021 and 725 in 2022) were included. From 2021 to 2022, nearly 80% of participants were aged <40 years old. More than 95% were female and nearly 60% were married. Of the respondents, the majority (67.9% in 2021 and 65.8% in 2022) of respondents were senior nurses, and nearly 70% of respondents needed to work in shifts. More than 90% of respondents were female (95.1% in 2021 and 95.9% in 2022). On average, more than half of the respondents had junior titles (67.9% in 2021 and 65.8% in 2022). Regarding their work experience, more than 75% of all respondents had less than 15 years of working experience. There were no statistical differences in the sociological characteristics in both the preintervention and postintervention periods between the treatment and control groups, indicating that the two groups were comparable ($p > 0.05$).

Table 2 shows the proportions of PVV exposure in the treatment and control groups in 2020 and 2021. Within the treatment group, the incidence of physical PVV significantly decreased from 13.8% in 2020 to 2.0% in 2021 ($\chi^2 = 18.16$, $p < 0.05$) and psychological PVV decreased from 35.5% in 2020 to 28.6% in 2021 ($\chi^2 = 1.89$, $p > 0.05$). In the control group, the incidence of physical PVV increased from 0.6% in 2020 to 2.7% in 2021 ($\chi^2 = 6.27$, $p < 0.05$), and psychological PVV decreased from 30.6% in 2020 to 23.0% in 2021 ($\chi^2 = 7.46$, $p < 0.05$). These results support the effect of safety and security equipment on nurses' exposure to physical PVV. However, further analysis is

**Table 1** Characteristics of the treatment and control group over 2020 and 2021, no. (%)

| Variable | Control group (n=1006) | | Treatment group (n=351) | | Overall (n=1357) | |
|---|---|---|---|---|---|---|
| | 2020 (n=480) | 2021 (n=526) | 2020 (n=152) | 2021 (n=199) | 2020 (n=632) | 2021 (n=725) |
| Age, years | | | | | | |
| ≤30 | 186 (38.8) | 243 (46.2) | 58 (38.2) | 70 (35.2) | 244 (38.6) | 313 (43.2) |
| 31–40 | 207 (43.1) | 199 (37.8) | 68 (44.7) | 89 (44.7) | 275 (43.5) | 288 (39.7) |
| >40 | 87 (18.1) | 84 (16.0) | 26 (17.1) | 40 (20.1) | 113 (17.9) | 124 (17.1) |
| Gender | | | | | | |
| Female | 461 (96.0) | 507 (96.4) | 140 (92.1) | 188 (94.5) | 601 (95.1) | 695 (95.9) |
| Male | 19 (4.0) | 19 (3.6) | 12 (7.9) | 11 (5.5) | 31 (4.9) | 30 (4.1) |
| Marital status | | | | | | |
| Never married | 162 (33.8) | 207 (39.4) | 69 (45.4) | 74 (37.2) | 231 (36.6) | 281 (38.8) |
| Married | 311 (64.8) | 310 (58.9) | 82 (53.9) | 123 (61.8) | 393 (62.2) | 433 (59.7) |
| Divorced/widowed | 7 (1.5) | 9 (1.7) | 1 (0.7) | 2 (1.0) | 8 (1.3) | 11 (1.5) |
| Job position | | | | | | |
| Junior/senior nurse | 313 (65.2) | 360 (68.4) | 107 (70.4) | 117 (58.8) | 420 (67.9 ) | 477 (65.8 ) |
| Supervisor nurse | 160 (33.3) | 162 (30.8) | 44 (28.9) | 78 (39.2) | 204 (32.3 ) | 240 (33.1 ) |
| Co-chief nurse | 4 (0.8) | 4 (0.8) | 1 (0.7) | 4 (2.0) | 5 (0.8 ) | 8 (1.1 ) |
| Chief nurse | 3 (0.6) | 0 (0.0) | 0 (0.0) | 0 (0.0) | 3 (0.5 ) | 0 (0.0 ) |
| Work experience | | | | | | |
| <1 | 19 (4.0) | 26 (4.9) | 11 (7.2) | 10 (5.0) | 30 (4.7 ) | 36 (5.0 ) |
| 1–5 | 128 (26.7) | 145 (27.6) | 52 (34.2) | 61 (30.7) | 180 (28.5 ) | 206 (28.4 ) |
| 6–10 | 156 (32.5) | 158 (30.0) | 36 (23.7) | 41 (20.6) | 192 (30.4 ) | 199 (27.4 ) |
| 11–15 | 97 (20.2) | 122 (23.2) | 29 (19.1) | 43 (21.6) | 126 (19.9 ) | 165 (22.8 ) |
| 16–20 | 40 (8.3) | 39 (7.4) | 5 (3.3) | 11 (5.5) | 45 (7.1 ) | 50 (6.9 ) |
| Over 20 | 40 (8.3) | 36 (6.8) | 19 (12.5) | 33 (16.6) | 59 (9.3 ) | 69 (9.5 ) |
| Work in shifts | | | | | | |
| No | 143 (29.8) | 144 (27.4) | 51 (33.6) | 69 (34.7) | 194 (30.7 ) | 213 (29.4 ) |
| Yes | 337 (70.2) | 382 (72.6) | 101 (66.4) | 130 (65.3) | 438 (69.3 ) | 512 (70.6 ) |
| Work between 6:00 PM and 7:00 AM | | | | | | |
| No | 211 (44.0) | 227 (43.2) | 95 (62.5) | 133 (66.8) | 306 (48.4 ) | 360 (49.7 ) |
| Yes | 269 (56.0) | 299 (56.8) | 57 (37.5) | 66 (33.2) | 326 (51.6 ) | 365 (50.3 ) |
| Interacting with patients/clients | | | | | | |
| No | 49 (10.2) | 47 (8.1) | 39 (25.7) | 113 (20.6) | 88 (13.9 ) | 88 (12.1 ) |
| Yes | 431 (89.8) | 479 (91.9) | 41 (74.3) | 158 (79.4) | 544 (86.1 ) | 637 (87.9 ) |
| Number of staff present in the same work setting | | | | | | |
| 0 | 5 (1.0) | 4 (0.8) | 2 (1.3) | 4 (2.0) | 7 (1.1) | 8 (1.1) |
| 1–5 | 159 (33.1) | 212 (40.3) | 64 (42.1) | 73 (36.7) | 223 (35.3) | 285 (39.3) |
| 6–10 | 181 (37.7) | 183 (34.8) | 40 (26.3) | 63 (31.7) | 221 (35.0) | 246 (33.9) |
| 11–15 | 58 (12.1) | 57 (10.8) | 26 (17.1) | 22 (11.1) | 84 (13.3) | 79 (10.9) |
| Over 15 | 77 (16.0) | 70 (13.3) | 20 (13.2) | 37 (18.6) | 97 (15.3) | 107 (14.8) |

needed to rule out the effect of time variables to determine whether these results are statistically significant.

The results of the DID model are presented in table 3. The application of safety and security equipment was associated with a 13.9% point decrease in the percentage of physical PVV occurring in the treatment group within 1 year in comparison with the control group (β=−13.93, p=0.024). These results are consistent with the descriptive statistics presented in table 2. The coefficients of safety and security equipment were not significant in the DID models when psychological PVV incidence and overall PVV incidence were treated as dependent variables. The findings of this study indicate that safety and security equipment can be considered protective factors against physical PVV.

**Table 2** The proportion of exposure to PVV in treatment and control groups over 2020 and 2021

| Type of violence | Year | Control group (n=1006) | Treatment group (n=351) | $\chi^2$ | P value |
|---|---|---|---|---|---|
| Physical PVV no. (%) | 2020 | 3 (0.6) | 21 (13.8) | 54.98 | 0.000 |
| | 2021 | 14 (2.7) | 4 (2.0) | 0.25 | 0.791 |
| $\chi^2$ (2021–2020) | | 6.27 | 18.16 | | |
| P value | | 0.014 | 0.000 | | |
| Psychological PVV no. (%) | 2020 | 147 (30.6) | 54 (35.5) | 1.28 | 0.272 |
| | 2021 | 121 (23.0) | 57 (28.6) | 2.48 | 0.122 |
| $\chi^2$ (2021–2020) | | 7.46 | 1.89 | | |
| P value | | 0.007 | 0.203 | | |
| Overall PVV no. (%) | 2020 | 147 (30.6) | 58 (38.2) | 2.99 | 0.091 |
| | 2021 | 124 (23.6) | 57 (28.6) | 1.98 | 0.178 |
| $\chi^2$ (2021–2020) | | 6.34 | 3.54 | | |
| P value | | 0.013 | 0.067 | | |

PVV, patient and visitor violence.

Several potential confounding factors may have affected these results. A logit model was used to estimate whether confounding factors influenced the incidence of PVV. The results are shown in table 4. Confounding factors had no significant effect on the incidence of physical PVV, indicating that the DID models were robust. Overall, the use of safety and security equipment can significantly reduce the incidence of physical PVV.

Note: No significant differences were observed in the baseline characteristics of nurses' personal and workplace settings between the treatment and comparison groups.

## DISCUSSION

In this study, we used a DID design to assess the effects of safety and security equipment on PVV incidence among nurses. This study has several important findings. We found that the implementation of safety and security equipment reduced the incidence of physical PVV in nurses significantly in academic years from 2020 (pretreatment) to 2021 (post-treatment). This demonstrates that the safety and security equipment could protect the nurses from the physical PVV at 1-year follow-up in China, supporting the necessary preventive and protective measures taken for employers to minimise occupational risks to HWs, especially during the period of COVID-19.[24] Given the disproportionate physical and psychological PVV against nurses, the reduction of risk through safety and security equipment, both at the individual and organisational levels, may have provided a particularly salient sense of relief only for physical PVV.

### The occurrence of PVV in the epidemic

Our findings reveal a lower prevalence of physical PVV (1.77% in 2020 and 1.33% in 2021) and a higher prevalence of psychological PVV (14.81% in 2020 and 13.12% in 2021) attacks on nurses related to COVID-19. These results are consistent with a previous study conducted during the pandemic, which reported similar rates of PVV among nurses in China (15.8% in psychological PVV and 8.4% in physical PVV).[25] However, it is worth noting that the prevalence of PVV found in the current study appears to be lower than that observed in other countries.[26] Evidence suggests that risk factors for PVV in the COVID-19 pandemic include high workload, crowded work environment and increased stress level brought by

**Table 3** Result from the DID analysis on the use of safety and security equipment

| Variable | Physical PVV | Psychological PVV | Overall PVV |
|---|---|---|---|
| Safety and security equipment (Std. error) | **−13.93*** **(4.893)** | 6.23 (9.074) | 0.88 (11.11) |
| Constant (Std. error) | 6.73† (0.950) | 27.67† (2.961) | 29.44† (3.182) |
| Observations | 18 | 18 | 18 |
| R-squared | 0.355 | 0.035 | 0.004 |

*p<0.05
†p<0.01
PVV, patient and visitor violence;

**Table 4** Logit regression results for control variables

| Variable | Coef. | Std. err. | Z | P | 95% CI | |
|---|---|---|---|---|---|---|
| Age | 0.245 | 0.221 | 1.11 | 0.267 | −0.187 | 0.677 |
| Marital status | 0.089 | 0.109 | 0.81 | 0.420 | −0.126 | 0.302 |
| Job position | 0.996 | 0.793 | 1.25 | 0.209 | −0.559 | 2.550 |
| Work experience | −0.356 | 0.280 | −1.27 | 0.203 | −0.905 | 0.192 |
| Work in shift | 0.371 | 0.331 | 1.12 | 0.262 | −0.277 | 1.019 |
| Work between 6:00 PM and 7:00 AM | −0.279 | 0.199 | −1.40 | 0.162 | −0.670 | 0.111 |
| Interacting with patients/clients | −0.104 | 0.146 | −0.71 | 0.476 | −0.389 | 0.182 |
| Number of staff in the same work setting | −0.137 | 0.190 | −0.72 | 0.471 | −0.509 | 0.235 |

the patient and visitor within healthcare settings.[27] In this context, visible safety and security measures, such as the presence of security personnel, surveillance cameras and restricted access to certain areas, can act as deterrents to potential aggressive behaviour of patients and visitors. While these measures are primarily intended to enhance physical safety by preventing violence, their presence can also contribute to a sense of security and comfort in healthcare workers.[28] These measures might have reduced the likelihood of violent incidents stemming from hazardous weapons, offering a sense of security that could deter potential aggressive actions.[29]

### Effect of the safety and security equipment on PVV

In this study, we made a unique contribution by quantifying the treatment effects of safety and security equipment on the incidence of PVV in public hospitals. Our findings provide evidence that implementing safety and security equipment in these hospitals led to a significant reduction of 13.9% points in the incidence of physical PVV. These findings align with the previous research on security strategies that effectively reduce WPV.[30 31] Existing evidence suggests that several risk factors contribute to nurses experiencing PVV, including unsafe and crowded work environments, easy public access to the facility and lack of security measures.[32] Safety and security equipment plays a crucial role in identifying potential threats and ensuring self-protection. One key reason is their deterrent effect. The security measures implemented in hospitals, such as the use of metal detectors at entrances and restricting the carrying of knives and other dangerous materials inside medical institutions, protect the personal safety of nurses. Installing functioning surveillance cameras and advanced access control systems in high-risk areas like EDs and waiting rooms, creating a sense of risk and discouraging potential perpetrators from engaging in violent behaviour, and making fewer areas accessible to the public, all contribute to personal safety of nurses.[33] In addition, installing safety and security equipment helps create a secure environment in the workplace, promoting a culture of safety and respect. When nurses feel protected and confident in their surroundings, the likelihood of PVV decreases. Security measures send a clear message that the organisation prioritises the well-being of its employees and has taken steps to ensure their safety. This fosters a positive work atmosphere, reducing tensions and conflicts that could escalate into violence.

However, the increase in physical PVV in the control group, where security measures were not implemented, raises concerns about the safety of nurses and staff members in those departments. The absence of security measures could potentially expose them to a higher risk of violence. It is crucial to consider additional factors that may contribute to the varying outcomes among groups, which means the presence of specific risk factors in inpatient department. This may be attributed to the ward's environment and patient population, such as insufficient human resources, patient demographics, the acuity of illness, overcrowding and staffing levels that can all impact the occurrence of PVV.[34]

Practically, while the safety and security equipment appears effective in curbing physical PVV, it provides limited affection in preventing psychological PVV. The physical PVV could be more visibly deterred by security equipment and be as easily controlled by physical security measures.[35] For example, the machine recognises that there is a danger, and security personnel can expel the relevant person from the hospital. However, although recognising psychological PVV through smart devices like facial recognition systems is possible, the absence of design, implement and monitor a workplace policy to prevent and combat psychological PVV makes it challenging to implement necessary follow-up measures. When a system detects a patient or visitor verbally abusing HWs, there is an absence of procedures to prohibit discrimination and harassment and stigma.[36] Based on this, a broader strategy should be considered. Initiatives such as establishing a management and coordination mechanism for workplace policy to prevent and combat violence, raising awareness through various initiatives, providing training, managing human resources effectively and improving communication can be beneficial.[37]

### Safety and security equipment in public hospitals

Safety and security equipment as environmental prevention strategies attracted more and more attention. Notably, the implementation of safety and security equipment rose sharply in 2022 in public hospitals in China. Most tertiary

hospitals in Beijing have established safety and security equipment by installing security cameras, biometric identification, smart cards and alarm buttons, providing security staff with walkie-talkies and increasing the number of metal detectors and security channels, perhaps because of the revision of the Promotion of Basic Medical Care and Health Law of the People's Republic of China. The joint efforts of the Chinese government, society and individuals have promoted legislation that protects healthcare workers and the development of authoritative WPV guidelines.[38] The Regulations on Hospital Safety Order Management of Beijing was put into effect in July 2020, stipulating that 'hospitals shall establish a safety inspection system and conduct safety inspection at the entrance of hospitals or key areas as needed to strictly prevent prohibited and restricted items from entering hospitals'.[39]

The results emphasise the importance of considering the unique characteristics and requirements of different departments when formulating security strategies. The control group and treatment group may face different work environments and potential risks, necessitating tailored security measures. The OD and ED are often high-traffic areas with a significant influx of patients and visitors. Implementing security measures can minimise crowding and reduce the risk of conflicts or acts of violence.[40] Inpatient wards necessitate a distinct security approach due to the potential prolonged presence of violence from patients and their families. Security measures should focus on controlling access to these areas and ensuring the safety of both patients and healthcare professionals. Implementing access control systems with visitor management protocols, surveillance cameras in corridors and panic buttons for staff members can help prevent unauthorised access and provide a rapid response in case of emergencies. Support and administrative areas may not directly involve patient care; they still require security measures to safeguard the well-being of healthcare professionals. Implementing access control systems, Closed Circuit Television cameras and staff identification protocols can help create a secure environment in these areas. [37] In addition, our outcomes from panel data of departments will not just impact the incidence of PVV on the part of the perpetrators but will also raise awareness of violence prevention and control among administrators at the managerial level from the long term, prompting them to put in place more violence-prevention measures and contributing to overall hospital improvement.

### Policy implications

In summary, the implementation of safety and security equipment in the treatment group led to a significant reduction in physical PVV incidents. This highlights the effectiveness of these measures in creating a safer environment for nurses. However, the increase in PVV in the control group without safety and security equipment underscores the importance of extending security protocols to all areas of the hospital. A lot of work remains to be done to promote safety and security equipment and

uptake in China; specifically, more effective legislation and policies are required from the perspective of the organisation and environment to reduce the occurrence of PVV in hospitals. It is crucial for hospital administrators to carefully assess the unique needs of each department and implement comprehensive security measures to ensure the safety and well-being of healthcare staff throughout the entire facility.[41]

### Limitations

This study has some limitations. First, this article selected nurses as research subjects, which limits our ability to generalise the findings to all HWs. However, the participants from various departments were all included in the study during the COVID-19 pandemic, which could also provide evidence of PVV prevention for other HWs during the epidemic period to some extent. Second, we collected data over the last two years and focused on the principal short-term impact of the current conditions; long-term effects need to be further monitored and evaluated. Third, the same as the previous studies, the recall bias is inevitable. However, consistent sampling methods and electronic questionnaires were used in the first-year and second-year surveys making the data collection convenient and safe, which could reduce the reporting bias to some extent. Fourth, our exclusive focus is on a single hospital, yet it is essential to note that this hospital is a large medical group comprising three public tertiary teaching hospitals. Finally, the safety and security equipment may lead to spillover effects, which could potentially lead to contamination of the control group with the intervention effects, thereby possibly underestimating the intervention effect of the safety and security equipment on PVV.

### CONCLUSIONS

Our study demonstrated that organisational and environmental improvement approaches that install safety and security equipment are effective ways to decrease physical PVV in China. These findings suggest that tertiary hospitals and other healthcare institutions can improve nurses' safety by incorporating safety and security equipment into their environmental improvement designs. We hope that these measures can be achieved to enact a 'zero' tolerance policy and that patient-physician trust can be rebuilt in China in the coming decade. Maintenance of a sound and safe order of medical services helps to form a harmonious doctor-patient relationship.

**Acknowledgements** We are very grateful for the assistance of the workplace violence research group of Professor Ju Huang at the Institute of Medical Information, Chinese Academy of Medical Sciences and Peking Union Medical College. Finally, we would also like to acknowledge the contributions of other staff members of our hospital who participated in the study.

**Contributors** JH organised, designed and conducted the study since 2021. YH, DZ and JX conducted the survey in 2022. YH organised the database, conducted the literature search and wrote the first draft of the manuscript. YH, JX and CZ performed the statistical analysis. JH modified the manuscript. X-mL conducted

critical guidance. All authors made significant contributions to the manuscript, read and approved the final version. Guarantor:YH

**Funding** This research was funded by the National Natural Science Foundation of China (Project Identification Code: 71804192). The funder had no influence on study design, data collection and analysis, report writing or publication of the article.

**Competing interests** The authors declare that the research was conducted in the absence of any commercial or financial relationships that could be construed as a potential conflict of interest.

**Patient and public involvement** Patients and/or the public were not involved in the design, or conduct, or reporting, or dissemination plans of this research.

**Patient consent for publication** Not applicable.

**Ethics approval** This study involves human participants and was approved by the Ethical Review Committee of the Chinese Academy of Medical Sciences (IMICAMS/8/22/HREC). Participants gave informed consent to participate in the study before taking part.

**Provenance and peer review** Not commissioned; externally peer reviewed.

**Data availability statement** No data are available. The raw data supporting the conclusions of this article will be made available by the authors, without undue reservation.

**ORCID iDs**
Yanzhen Hu http://orcid.org/0000-0003-4974-9664
Ju Huang http://orcid.org/0000-0002-7810-4550
Jinghua Xia http://orcid.org/0000-0003-2701-4501

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
