## [Reviewer comments · BMJ Open]

ARTICLE DETAILS

TITLE (PROVISIONAL)	Effect of Safety and Security Equipment on Patient and Visitor Violence toward Nurses in multiple public hospitals of China during COVID-19 pandemic : A retrospective, difference-in-difference analysis
AUTHORS	Hu, Yanzhen; Huang, Ju; Zhao, Dan; Zhang, Cheng; Xia, Jinghua; Lu, Xue-mei

VERSION 1 – REVIEW

REVIEWER	Henriques, Helga Rafael Escola Superior de Enfermagem de Lisboa
REVIEW RETURNED	08-Oct-2023

GENERAL COMMENTS	Thank you very much for the opportunity to analyze this article. This is a well-written and meticulously organized article characterized by the implementation of robust research methods. The authors have demonstrated a commendable proficiency in articulating their study, making it accessible and coherent for readers. The clarity in presentation is particularly evident in the meticulous organization of sections, such as the introduction, methods, results, discussion, and limitations. The robustness of the research methodology employed in the study enhances its credibility. A Difference-in-Differences (DID) design is commendable for its ability to establish causal relationships, offering a robust foundation for concluding the impact of safety and security equipment on reducing patient and visitor violence (PVV) among nurses. This methodological choice strengthens the validity of the study's findings and contributes to its overall reliability. The introduction is well-structured, providing a comprehensive overview of the problem of workplace violence against health workers, with a specific focus on nurses during the COVID-19 pandemic in China. It combines global perspectives, specific challenges faced in China, government responses, and existing research gaps to justify the need for the study. While the introduction highlights the lack of evidence on the effectiveness of safety measures in preventing violence, it might be beneficial to briefly discuss what is expected to be found in the study results and how this may contribute to existing knowledge. The methods section is well-structured and clearly explains the study design, sampling strategy, data collection, and statistical analysis. The use of appropriate statistical methods and consideration of potential confounders contribute to the robustness of the research design. However, there are a few suggestions for potential improvement:
--

	 - Provide a brief justification or rationale for selecting specific departments, especially why the outpatient, emergency, and technical services were chosen as the treatment group. - Offer more details on safety and security equipment types and specifics. - Consider including a flow diagram depicting the participant selection process. In discussing the incidence of PVV related to COVID-19, consider discussing the impact of the pandemic on stress levels, patient, and visitor frustration, and how safety and security measures may have played a role. Once the incidence of psychological WPV was not significantly affected by security equipment, discuss practical implications and possible reasons for this outcome. As noted by the authors, the study has limitations regarding generalizability due to its focus on a single tertiary hospital. It is suggested that future research. The authors could explore more specific considerations or propose research avenues that delve into the long-term effects of the interventions or conditions under examination post-pandemic. Review sentence formatting line 235
--	--

REVIEWER	Lin , Jin-Ding Mackay Medical College, Institute of Long Term Care
REVIEW RETURNED	10-Oct-2023

GENERAL COMMENTS	This study collected data from a single study setting (a large public hospital)and will provide limited knowledge on the study issue. The comparing group is lacking, and the validity and relaiability of data needed to evaluate in detail.
---

VERSION 1 – AUTHOR RESPONSE

Reviewer 1 (Comments for the Author):

Comment 1: While the introduction highlights the lack of evidence on the effectiveness of safety measures in preventing violence, it might be beneficial to briefly discuss what is expected to be found in the study results and how this may contribute to existing knowledge.

Response 1: Accepted and revised (Page 3 lines 105-112). We have modified the introduction and provided a clear justification for what is expected to be found in the study results and how this may contribute to existing knowledge.

The PVV incidents toward nurses during COVID-19 pandemic, and its causal correlations with implementation of safety and security measures are expected to be found in the study. In addition, the DID model used in this study could provide quantitative data to existing knowledge, quantifying the reduction rate in PVV with the implementation of safety and security measures. It offers a more objective data reference for the addition of new security measures in the next phase, furthering the goal of zero tolerance for physical violence. Such findings offer valuable guidance for hospitals in other developing countries confronting similar situation. Moreover, our research augments evidence-based practices by providing empirical data on the efficacy of safety equipment.

Comment 2: Provide a brief justification or rationale for selecting specific departments, especially why the outpatient, emergency, and technical services were chosen as the treatment group.

Response 2: Accepted and revised (Page 6 lines 135-146). We have provided a brief justification for selecting specific departments.

According to the Medicine and Health Promotion Law, based on the epidemiological characters of PVV, the hospital administration strategically selected three departments- OD, ED, and TSD-for the implementation of safety and security measures. Therefore, the participants from departments installed the safety and security equipment in the end of 2020 were treatment groups (OD, ED, and TSD), and the participants from departments with no safety and security equipment were control groups: general medicine, general surgery, operating room, intensive care, specialized unit, and management department. The baseline data for 2020 were collected in January 2021, and we followed-up with the selected departments that previously consented to participate in the panel for data of 2021 in January 2022. Ultimately, a total of 632 eligible nurses participated in the survey 2021 to report the PVV in 2020, of whom 152 were from the treatment group and 480 were from the control group. In survey 2022, 725 eligible nurses participated to report the PVV in 2021, of whom 199 were from the treatment group and 526 from the control group.

Comment 3: Offer more details on safety and security equipment types and specifics.

Response 3: Accepted and revised (Page 5 lines 157-166). We added the safety and security equipment types and specifics in methods.

Metal detectors (SM-SG-EDD-5000, Sinomis) and security channel (ISD-SC6550D-4YH, Sinomis) were set up at entrances in the outpatient building and emergence building to screen individuals for any potentially harmful items that could jeopardize the safety of medical personnel. Security cameras (IVS3800XF36 V5, Huawei) installed at multiple corners to covering the entire building, providing constant surveillance and aiding in the identification of potential risks or incidents; Alarm buttons (NNV-30, LonBon) installed under the triage table in ED and reception in OD, allowed immediate and discreet alerting of security personnel in case of emergencies or threatening situations. About 70 security personnel, each with a walkie-talkie (MTP3150, Xinzhiheng), facilitated swift communication and coordination among them, enabling quick responses to any security issues.

Comment 4: Consider including a flow diagram depicting the participant selection process.

Response 4: Accepted and revised.

Comment 5: In discussing the incidence of PVV related to COVID-19, consider discussing the impact of the pandemic on stress levels, patient, and visitor frustration, and how safety and security measures may have played a role.

Response 5: Accepted and revised (Page 10 lines 268-276).

Evidence suggests that risk factors for PVV in the COVID-19 pandemic include high workload, crowded work environment, and increased stress level brought by the patient, and visitor within healthcare settings. In this context, visible safety and security measures, such as the presence of security personnel, surveillance cameras, and restricted access to certain areas, can act as deterrents to potential aggressive behavior to patient and visitor. While these measures are primarily intended to enhance physical safety by preventing violence, their presence can also contribute to a sense of security and comfort in healthcare workers. These measures might have reduced the likelihood of violent incidents stemming from hazardous weapons, offering a sense of security that could deter potential aggressive actions.

Comment 6: Once the incidence of psychological WPV was not significantly affected by security equipment, discuss practical implications and possible reasons for this outcome.

Response 6: Accepted and revised (Page 11 lines 304-316). We added some details about practical implications and possible reasons for this outcome.

Practically, while the safety and security equipment appear effective in curbing physical PVV, it provides limited affection in preventing psychological PVV. The physical PVV could be more visibly

deterred by security equipment, and be as easily controlled by physical security measures. For example, the machine recognizes that there is a danger, and security personnel can expel the relevant person from the hospital. However, although recognizing psychological PVV through smart devices like facial recognition systems is possible, the absence of design, implement and monitor a workplace policy to prevent and combat psychological PVV makes it challenging to implement necessary follow-up measures. When a system detects a patient or visitor verbally abusing HWs, there is an absence of procedures to prohibit discrimination and harassment and stigma. Based on this, a broader strategy should be considered, Initiatives such as establishing a management and coordination mechanism for workplace policy to prevent and combat violence, raising awareness through various initiatives, providing training, managing human resources effectively, and improving communication can be beneficial.

Comment 7: It is suggested that future research. The authors could explore more specific considerations or propose research avenues that delve into the long-term effects of the interventions or conditions under examination post-pandemic.

Response 7: We greatly appreciate your advice, and our team will seriously consider your suggestions in our future research.

Comment 8: Review sentence formatting line 235

Response 8: Accepted and revised. (Page 10 lines 261)

Reviewer 2 (Comments for the Author):

Thank you for taking the time to engage with my research paper. I appreciate your interest and your comments. I wanted to clarify the two points you raised in the study.

Comment 1: This study collected data from a single study setting (a large public hospital) and will provide limited knowledge on the study issue.

Response 1: While this study exclusively focuses on a hospital medical group, the independent operation and varied geographic distribution across three campuses strengthen its representative.

Comment 2: The comparing group is lacking, and the validity and reliability of data needed to evaluate in detail.

Response 2: The paper does indeed incorporate a control group within the study design. The DID method involves comparing the changes over time between two groups: the intervention group and the control group. The intervention group received the specific treatment or intervention, while the control group did not. In our study, the participants from departments installed the safety and security equipment in the end of 2020 were treatment groups (OD, ED, and TSD), and the participants from departments with no safety and security equipment were control groups: general medicine, general surgery, operating room, intensive care, specialized unit, and management department. The baseline data for 2020 were collected in January 2021, and we followed-up with the selected departments that previously consented to participate in the panel for data of 2021 in January 2022. Ultimately, a total of 632 eligible nurses participated in the survey 2021 to report the PVV in 2020, of whom 152 were from the treatment group and 480 were from the control group. In survey 2022, 725 eligible nurses participated to report the PVV in 2021, of whom 199 were from the treatment group and 526 from the control group.

VERSION 2 – REVIEW

REVIEWER	Henriques, Helga Rafael Escola Superior de Enfermagem de Lisboa
REVIEW RETURNED	01-Jan-2024

GENERAL COMMENTS

The authors conscientiously addressed the information gaps, improving the article's quality. I recommend the article for publication, underscoring the authors' commitment to delivering a comprehensive and noteworthy contribution to the field.